# The association between self-reported stress and cardiovascular measures in daily life: A systematic review

Thomas Vaessen[1,2]ᴑ*, Aki Rintala[1,3]ᴑ, Natalya Otsabryk[1], Wolfgang Viechtbauer[1,4], Martien Wampers[1,5], Stephan Claes[2,5], Inez Myin-Germeys[1]

1 Center for Contextual Psychiatry, Department of Neurosciences, KU Leuven, Leuven, Belgium, 2 Center for Mind-Body Research, Department of Neurosciences, KU Leuven, Leuven, Belgium, 3 Faculty of Social Services and Health Care, LAB University of Applied Sciences, Lahti, Finland, 4 School for Mental Health and Neuroscience, Department of Psychiatry and Neuropsychology, Maastricht University, Maastricht, The Netherlands, 5 University Psychiatric Center KU Leuven, KU Leuven-University of Leuven, Leuven, Belgium

ᴑ These authors contributed equally to this work.
* thomas.vaessen@kuleuven.be

**Data Availability Statement:** All relevant data are within the paper and its Supporting information files. More detailed extraction files are available from the corresponding author.

## Abstract

### Background

Stress plays an important role in the development of mental illness, and an increasing number of studies is trying to detect moments of perceived stress in everyday life based on physiological data gathered using ambulatory devices. However, based on laboratory studies, there is only modest evidence for a relationship between self-reported stress and physiological ambulatory measures. This descriptive systematic review evaluates the evidence for studies investigating an association between self-reported stress and physiological measures under daily life conditions.

### Methods

Three databases were searched for articles assessing an association between self-reported stress and cardiovascular and skin conductance measures simultaneously over the course of at least a day.

### Results

We reviewed findings of 36 studies investigating an association between self-reported stress and cardiovascular measures with overall 135 analyses of associations between self-reported stress and cardiovascular measures. Overall, 35% of all analyses showed a significant or marginally significant association in the expected direction. The most consistent results were found for perceived stress, high-arousal negative affect scales, and event-related self-reported stress measures, and for frequency-domain heart rate variability physiological measures. There was much heterogeneity in measures and methods.

**Funding:** This study was supported by the Remote Assessment of Disease and Relapse – Central Nervous System (RADAR-CNS) research programme from the Innovative Medicines Initiative 2 Joint Undertaking under a grant agreement number 115902. This Joint Undertaking receives support from the European Union's Horizon 2020 research and innovation programme and the European Federation of Pharmaceutical Industries and Associations (EFPIA). Inez Myin-Germeys and Thomas Vaessen were funded by the Fonds voor Wetenschappelijk Onderzoek (FWO) Odysseus grant (GOF8416N). Thomas Vaessen was supported by a FWO postdoc grant (1243620N) and by a Horizon 2020 grant (ZL384206-MOODSTRATIFICATION).

**Competing interests:** The authors have declared that no competing interests exist.

## Conclusion

These findings confirm that daily-life stress-dynamics are complex and require a better understanding. Choices in design and measurement seem to play a role. We provide some guidance for future studies.

## 1 Introduction

Stress is one of the largest environmental risk factors for mental illness. According to diathesis-stress models, prolonged exposure to stressors can lead to severe mental illness in vulnerable individuals [1–3]. Simultaneously, prolonged exposure to stressors sensitizes the stress system, resulting in altered affective reactivity to relatively minor stressors, such as daily hassles [4]. An individual's affective reactivity to these minor stressors is therefore thought to reflect an underlying risk of developing mental illness. In line with this theory, diary studies have indicated that increased affective reactivity to daily hassles mediates the effect of childhood adversity on psychopathology [5–7]. Increased affective reactivity to daily stressors has been associated with a number of mental illnesses [8–12], making it an important indicator of mental health.

Daily stress can be measured using ambulatory assessment methods. To date, the majority of studies investigating everyday stress have done so using structured diary techniques (i.e., experience sampling methodology [ESM]; ecological momentary assessment [EMA]) to assess the subjective experience of being stressed and its effects on affective states. Typically, study participants are provided with a device that sends a signal each time a diary entry is required. Such diary entries consist of questions on momentary experiences, contexts, and appraisals, providing insight into the participants' daily lives while keeping recall bias low [13, 14]. Using these diaries, self-reports on experienced stress levels (henceforth referred to as self-reported stress) have been studied in a variety of ways, in part depending on the underlying theory [15, 16]. Stress as a concept has many definitions. The most prominent theories posit that stress is bodily strain in response to demand [17] or an allostatic reaction to a perceived threat [18] that occurs when a situation is appraised as more challenging, unpleasant, and important than the individual can cope with [19] or when perceived demands are greater than perceived control over the situation [20]. Although different, these definitions all seem to assume that stress depends on an individual's perception of a given situation. Circularly, this means an individual is under stress when they perceive a situation as stressful (i.e. demanding, threatening, etc.). As a result, self-reported stress has been operationalized as appraisals, perceived stress, or affective distress (or negative affect [NA]) in countless different ways [15, 16]. Two reviews have investigated how these studies assessed stress in daily life and found much heterogeneity in measures [15, 16]. This heterogeneity is most likely a reflection of the variety in theoretical definitions, terminologies, and approaches.

Due to advances in mobile technology, the last decade has seen a steep increase in the number of studies assessing the autonomic nervous system (ANS) response to daily stressors using wearable devices [15]. An advantage of these ambulatory remote monitoring methods is that they do not require immediate action from the participants in order to collect data; many wearable sensors can gather data passively throughout the study period. ANS measures collected using wearable sensors include blood pressure, heart rate, and skin conductance, which have all been positively associated with psychosocial stress [21, 22]. Heart rate variability (HRV) is another physiological measure that has been linked to stress, typically in the form of a negative relationship. Considering the relevance of stress reactivity for mental health, being able to detect instances of stress reactivity that signal psychopathological vulnerability through

passively monitored ANS markers could potentially have a large impact on early intervention strategies in mental healthcare.

However, none of these measures are stress-specific, which reveals the method's Achilles' heel; when an experiential perspective is lacking, there is no certainty that changes in physiology reflect instances of acute stress. Yet, even combining daily life remote monitoring of physiology and an ESM assessment on self-reported stress may not provide the answer. In fact, over the years, several studies have tried to predict self-reported stress based on wearable sensor data, with varying levels of success [23]. A systematic review on reactivity to a standardized psychosocial stress task under laboratory conditions reported that only 25% of the studies they reviewed found an association between self-reported and cardiovascular measures of stress [24]. Moreover, suppressing the ANS [25], or both the ANS and the endocrine system [26] did not affect stress reports during a psychosocial stress task, begging the question whether these systems are associated at all. Still, theoretical frameworks assume this coherence between self-reported stress and physiological measures exists [27, 28]. Over the years, several studies have combined ESM and daily life remote monitoring of physiology when investigating the stress response in daily life, and have looked at associations between both types of measures. No systematic review to date, however, has compared these studies and their findings to evaluate the evidence that they assess the same underlying process. Moreover, much like the heterogeneity in self-report measures, several different physiological variables have been used to capture the stress response [15] and it is unclear what their individual relationships to self-reported stress are. Similarly, little is known about how differences in stress reactivity observed in different study populations affect the relationship between daily-life self-reported and physiological measures of stress. The same goes for choices on study devices such as wearable sensors or diary equipment. Finally, the study protocol and its compliance can have an influence when stressful moments are not sampled frequently enough.

This systematic review has three main aims: First, we aim to identify *how* studies have investigated the broad concept of daily life stress using simultaneous ambulatory measures of self-reported stress and cardiovascular and skin conductance features. Second, we will review the evidence that these measures of self-reported stress and cardiovascular and skin conductance features are associated. Third, we will explore the influence of choices on self-reported stress measures, physiological measures, study population, study methods, and compliance, on these associations. To do so, we will systematically review all studies that have assessed daily stress using ambulatory methods and associated ratings of self-reported stress with cardiovascular and skin conductance measures associated with the stress response.

## 2 Materials and methods

### 2.1 Search strategy

A systematic literature search was conducted using three databases (S1 File): Comprehensive Biomedical Literature Database (EMBASE), Archive of Biomedical and Life Sciences Journal Literature (PubMed), and Web of Science (WOS) Core Collection. The search was performed for studies published until 6th June 2019. An updated search was conducted from the same databases on studies published between 7th June 2019 and 25th October 2020. Fig 1 presents the combined flowchart of the study selection.

Inclusion criteria were designed by the research team taking into account only studies investigating the association between self-reported stress and cardiovascular measures and skin conductance measures in daily life. Self-reported stress was defined as an item or a feature of all forms inquiring about the subjective experience of stress. As one of the aims of this review was to investigate how self-reported stress in daily life is assessed, we opted for a liberal

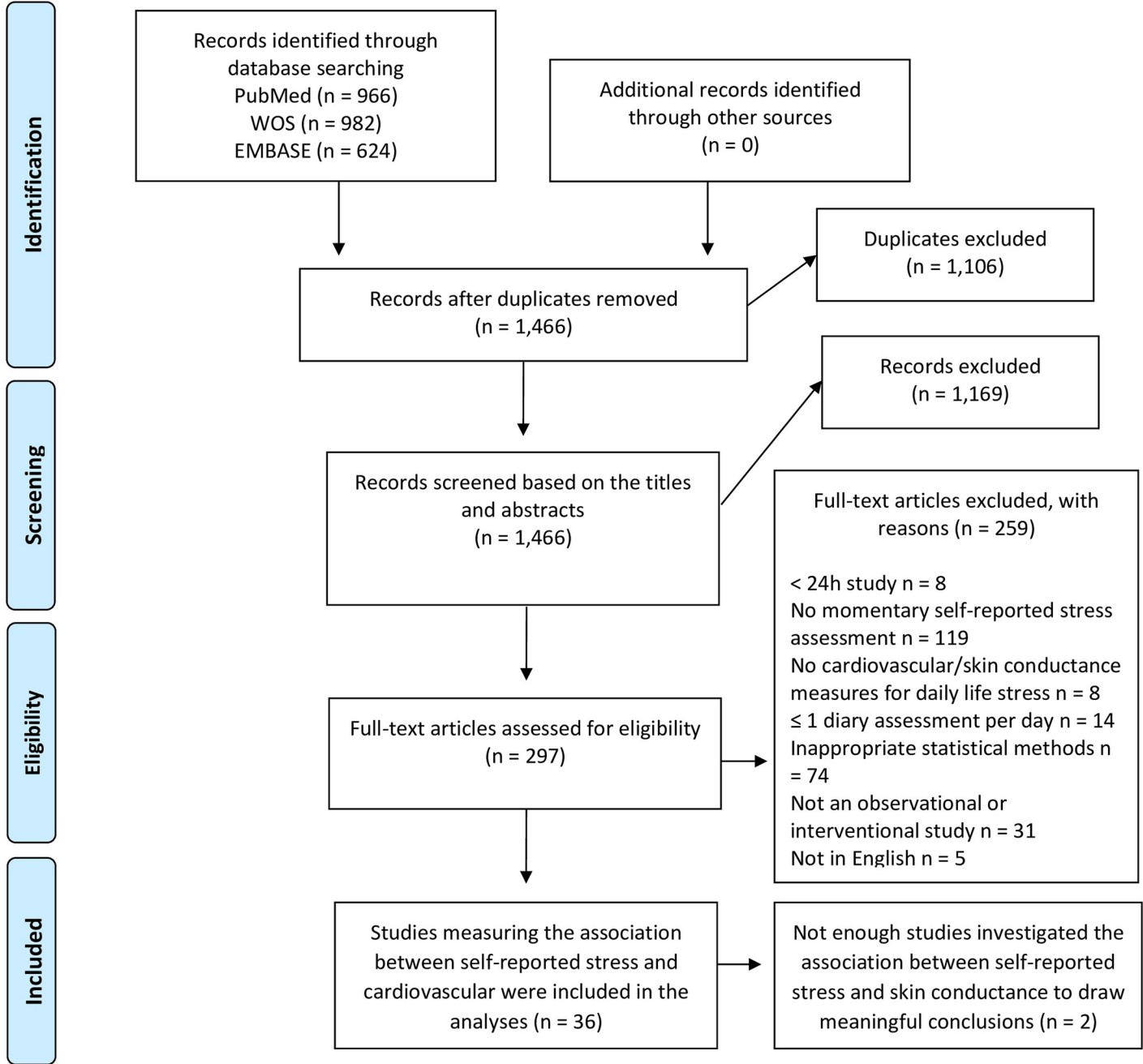

**Fig 1. Flow chart.**

approach in our literature search. Since some authors assess stress using NA scales (either including items on experience of stress or not), we also included search terms referring to affect. Studies that only assessed positive affect were excluded. Also, as some of the main stress theories may result in different operationalizations, we included several other terms. Specifically, appraisal theory states that stress may be a response to unpleasant or negative events (i.e. hassles) [29]. Other main theories use terms such as strain [20], demand [17], or threat [30].

Ambulatory cardiovascular and skin conductance measures were defined as any remote assessment measuring physiological data (e.g., ambulatory devices and/or wearable devices) in daily life. Both self-reported stress and cardiovascular and/or skin conductance measures in daily life needed to occur in a natural environment without the presence of a healthcare professional or a researcher, and assessing both measures simultaneously. For purposes of ecological validity and considering the dynamic process that is stress, we only included studies that reported more than one stress assessment per day (i.e., no end-of-day diary studies) and took into account the multilevel nature of the data in the statistical analyses. Also, our search strategy only included interventional and observational studies that were published in English. Studies publishing results from the same dataset were only included if they reported on different variables. Systematic reviews, discussion papers, study abstracts, qualitative studies, and study protocols were excluded.

A researcher (A.R.) performed the searches in the selected databases with the collaboration of the research team. Search terms included various self-reported and cardiovascular and skin conductance stress terms (e.g., "stress*", "distress", "threat", "cardiovascular", "skin conductance") and momentary or remote assessment protocol terms (e.g., "experience sampling", "momentary", "diary"). Search terms included either medical subject headings (MeSH) or keyword headings. The original search strategy is described in S1 File.

## 2.2 Data extraction

Two reviewers (A.R. and T.V.) independently screened the titles and abstracts of the studies in line with the Preferred Reporting for Systematic Reviews and Meta-analysis (PRISMA) guidelines using the defined search strategy [31, 32]. Next, relevant studies were independently evaluated for full-text assessment by two reviewers (A.R. and T.V.). An updated search was conducted with the same approach by two reviewers (A.R. and N.O.). A third reviewer (I.M.-G.) evaluated the studies in case of a disagreement. If needed, corresponding authors of the included studies were contacted for further information.

We extracted the following details from the included studies: publication year, study sample (study population, age, and sex), study methods (study length, frequency of the assessments, sampling design, and user devices), participant compliance to the protocol, self-reported stress (type of stress, number of stress items, description of the stress items, type of scales), cardiovascular and/or skin conductance measures (i.e., type of measure, variable used), and finally, the type of analysis used for associations of the stress assessments and covariates included, and its findings.

## 2.3 Methodological quality of the studies

Methodological quality of the included studies was evaluated using the Downs and Blacks Scale [33]. The checklist consists of 27 items and includes domains for study reporting (10 items), external validity (3 items), internal validity (bias and confounding) (13 items), and power (1 item) [33]. An item was scored 1 (Yes) if the criterion was fulfilled or 0 if inadequately reported, unable to determine, or not applicable. Overall quality rating per study was assessed using the corresponding quality levels as previously reported with a total possible score of 28 for randomized and 25 for non-randomized studies [34]: excellent (26–28); good (20–25); fair (15–19); and poor ($\leq$ 14). Study quality assessment was performed independently by one reviewer (AR), and in case of uncertainty, other members of the research team were consulted.

## 2.4 Statistical synthesis

Descriptive analyses were performed on all extracted variables. In case multiple studies reported on the same dataset, only the study with the largest sample size was considered for the descriptive analysis of the study sample (in case the sample sizes were identical, only the original publication was reported). Associations between self-reported stress and cardiovascular and/or skin conductance measures descriptively reported and linked with the type of measure (i.e., heart rate, heart rate variability, blood pressure, or skin conductance). If applicable, descriptive analyses were performed for associations based on study length and devices.

## 3 Results

The search identified overall 1,466 studies after the removal of duplicates. Screening of 297 full-text studies resulted in a total of 38 studies that fulfilled the inclusion criteria. A flow chart of the screening process is presented in Fig 1 and the extracted data are presented in Tables 1–4. Our search results identified only two studies reporting associations with self-reported stress and the level of skin conductance as a physiological variable [35, 36]. Since we considered this too few to draw any meaningful conclusions, this systematic review focuses only on studies investigating the association between self-reported stress and cardiovascular measures (36 studies).

### 3.1 Study characteristics

**3.1.1 Cardiovascular measures in daily life.** Our screening results identified three different types of cardiovascular measures: blood pressure, heart rate, and heart rate variability. Blood pressure was assessed in 19 reports [37–55] (Table 2), heart rate in 21 studies [37–39, 42, 43, 45–47, 51, 53, 56–66] (Table 3), and heart rate variability in 12 studies [56, 58, 59, 62, 64, 65, 67–72] (Table 4). Specifically, BP was measured as systolic (SBP) or diastolic (DBP) blood pressure, mean arterial pressure (MAP), or pulse pressure (PP). Heart rate (HR) was measured in beats per minute. Heart rate variability (HRV) was assessed using the time-domain measures rr interval, mean square of successive differences (MSSD), root mean square of successive differences (RMSSD), or frequency-domain measures of low frequency (LF-HRV) and high frequency (HF-HRV) heart rate variability. Some studies reported associations with the ratio of LF-HRV and HF-HRV (LF/HF ratio). Although the LF/HF ratio was suggested to reflect sympatho-vagal balance, it has been criticized as such [73]. As it is unclear to date what the LF/HF ratio represents, we will not report findings on this measure in this review.

**3.1.2 Self-reported stress measures.** As expected, the literature search revealed that self-reported stress was assessed in a large variety of ways (see Table 1 for an overview of all self-report measures used). In an attempt to categorize the approaches, we identified three main different ways in which researchers quantified self-reported stress:

1. The most common method in which self-reported stress was measured was NA, which was assessed either in the form of an average or sum score over several NA items or as a single item in 18 different studies. There was much heterogeneity in measures of NA. Most notably, some of the NA scales used in the studies reviewed here included low-arousal items such as "unhappy" [41, 50, 62, 69], "ashamed" or "embarrassed" [46], "sad" [41, 45, 62], or "lonely" [62], which arguably do not adequately reflect the subjective experience of stress. We therefore also looked specifically at analyses on high-arousal only measures of NA (i.e., scales that solely consist of high-arousal NA items).

2. Thirteen studies assessed self-reported stress directly through perceived stress, typically using only one item asking about current perceived feelings of stress. One study used a

**Table 1. Self-reported stress measures.**

| First author (year) | Scale name and construction (if applicable) and individual items | Scale | Number of items |
|---|---|---|---|
| **Perceived stress** | | | |
| Schilling (2020) | *"How stressed do you feel at the moment?"* | Likert (5-point) | 1 |
| Birk (2019) | *"Think about how you were feeling just before BP. How overwhelmed were you feeling?"* | VAS (0–100) | 1 |
| Krkovic (2018) | Average over [*"The situation was stressing me"*, *"I was able to control the situation"* (reversed), *"I was calm and relaxed"* (reversed), *"I was helpless in the situation"*, fear, sadness, anger, shame, guilt, and (un-)happiness][a] | Likert (10-point) | 10 |
| Dennis (2016) | Distressed | Likert (5-point) | 1 |
| Kennedy (2015) | Stressed | Likert (5-point) | 1 |
| Riediger (2014) | Nervous | Likert (7-point) | 1 |
| Pieper (2010) | Tense or restless | Likert (5-point) | 1 |
| Ebner-Priemer (2007) | *"How high was your distress just before the beep?"* | Likert (11-point) | 1 |
| Pollard (2007) | Stress level over the past hour | Likert (7-point) | 1 |
| Meininger (2004) | Stress | YES/NO | 1 |
| Bacon (2004) | Stressed | Likert (5-point) | 1 |
| Buckley (2004) | Stressor | YES/NO | 1 |
| Tsai (2003) | Tenseness | Likert (5-point) | 1 |
| **Negative affect** | | | |
| Schilling (2020) | Sum of five unspecified negative affect items | Likert (5-point) | 5 |
| Schwerdtfeger (2019) | Average over [upset, distressed, agitated, tense, nervous]* | Likert (7-point) | 5 |
| Dennis (2017, 2018) | Average over [irritated, annoyed, angry, distressed, upset, hostile, stressed]* | Likert (5-point) | 7 |
| Zawadzki (2016) | Valence of affect | Likert (7-point) | 1 |
| Edmondson (2015) | Cube root of *"Just before your BP was taken, how anxious/tense were you feeling?"** | VAS (0–100) | 1 |
| Lehman (2015) | Shame; anger*; embarrassment; anxiety* | VAS (0.1–10) | 4 |
| Kimhy (2014) | Highest rating of [anxiety, loneliness, irritation, sadness, happiness/relaxation (reversed)] | VAS (0–100) | 5 |
| Schwerdtfeger (2014) | Sum of [insecure, downhearted, anxious, ashamed, worried, dissatisfied] | Likert (6-point) | 6 |
| Friedmann (2013) | Average over [frustrated, angry, unhappy, nervous, rushed, irritable, sad, stressed] | VAS (0–100) | 8 |
| Ilies (2010) | Average over [upset, distressed, hostile]* | Likert (5-point) | 3 |
| Pieper (2010) | Angry or irritated*; sad or gloomy | Likert (5-point) | 3 |
| Bacon (2004) | Anger; sadness; tiredness | Likert (5-point) | 3 |
| Meiniger (2004) | Angry*; bored; irritable*; sad | YES/NO | 4 |
| Tsai (2003) | Annoyance* | Likert (5-point) | 1 |
| Carels (2000) | Sum of [tension, frustration, stress]* | Likert (5-point) | 3 |
| Picot (1999) | Angry*; anxious*; happy (inversed) | VAS (0–10) | 3 |
| Kamarck (1998) | Average over [sad, frustrated, stressed, upset] | Likert (4-point) | 4 |
| Sloan (1994) | Average over [happy (reversed), irritable, tense, Pressured] | Likert (7-point) | 4 |
| **Event-related stress** | | | |
| Pieper (2010, 2007) | Occurrence of minor stressful event in the past 60 minutes | YES/NO | 1 |
| Luecken (2009) | Occurrence of minor stressful event in the past 30 minutes | YES/NO | 1 |
| **Activity-related stress** | | | |
| Schmid (2020) | *Work-related emotional demands*: *"My work puts me in emotionally disturbing situations"* | Likert (7-point) | 1 |
| Thomas (2019)[b] Kamarck (2018)[b] | *Task Strain*: Control (average over [*"Could change activity if you chose to?"* and *"Choice in scheduling this activity?"*]) <4 (6-point Likert scale) Demand (average over [*"Required working hard?"*, *"Required working fast?"* and *"Juggling several tasks at once?"*]) > 3 (6-point Likert scale) | YES/NO | 2 + 3 |
| Johnston (2016) | *Work-related Stress*: Interaction between Demand [work fast, work hard, do too much, interrupted, enough time available] and Control [requiring a high level of skill, allowed a lot of say in what they did, allowed to make the main decisions about what they did] | YES/NO | 5 + 3 |
| Hawkley (2003) | *Activity-related stress and threat*: *"How stressful and threatening is the main thing you are doing?"* | Likert (5-point) | 1 |

*(Continued)*

**Table 1.** (Continued)

| First author (year) | Scale name and construction (if applicable) and individual items | Scale | Number of items |
|---|---|---|---|
| Hawkley (2003) | *Cognitive appraisal ratings*: Ratio of how demanding they found the main activity to the degree to which they felt capable of meeting the demands of the activity | Likert (5-point) | 1 |
| Kamarck (1998) | *Task Strain*: Interaction between Demand (average over ["*hard work*", "*fast work*", and "*juggling tasks*"]) and Control (average over ["*can change activity*" and "*chose activity*"]) | Likert (4-point) | 3 + 2 |
| **Social stress** | | | |
| Thomas (2019) [b] Kamarck (2018)[b] | *Social Conflict*: Average over ["*Someone in conflict with you*?" and "*Someone treated you badly*?"] | Likert (6-point) | 2 |
| Cornelius (2018) | *Social Interaction Pleasantness*: Person-mean centered pleasantness of the social interaction | VAS (0–10) | 1 |
| Lehman (2015) | *Subjective Social Evaluative Threat*: Average over z-transformed average over 4 items ["*I was focused on what others thought of me*", "*I felt like the center of attention*" (sample items; 5-point Likert scale) and "*I was worried about others' reactions to me*" (10-point VAS)] | Likert (5-point) VAS (0–10) | 3 |
| Kennedy (2015) | "*Someone hassling you*" | Likert (5-point) | 1 |
| Lehman (2010) | *Social-evaluative Threat*; "*I was worried about others' reactions to me*" | VAS (1–10) | 1 |
| Kamarck (1998) | *Social Conflict*: Average over [someone made unfair demands of you, interrupted you, judgmental or critical of you, ignored you, argued with you] | Likert (4-point) | 5 |

VAS: visual analogue scale.

[*] High-arousal negative affect measures.

[a] This study used a combination of perceived stress and NA items.

[b] these studies report on the same sample with similar analyses, we used the results from Thomas (2019) for all our purposes.

combination of perceived stress and NA measures to compute self-reported stress [63]. As this mixture might confound our results, we included this study only in the overall analyses and not in any of the sub-analyses on specific stress types.

3. All other types of self-reported stress were situational, inquiring about experienced stress related to current or recent situations or events. Specifically, three studies assessed associations with measures that involved recent stressful events, which we categorized as event-related stress. In both of these studies, participants were asked about the occurrence of a minor stressful event either in the past 60 or 30 minutes. Six studies reported findings on stress or strain related to a current or a recent task or activity, which we identified under the term activity-related stress, although each of the included studies opted for slightly different approaches (i.e., task strain, work-related stress, activity-related stress, and cognitive appraisal). Finally, seven studies included a measure of stress in social company or situations, which we clustered and viewed as social stress measures. However, each of these studies operationalized social stress differently (i.e., social conflict, feeling of annoyance, pleasantness of social interaction, and social-evaluative threat).

Scales that were used to measure self-reported stress were very heterogeneous; the most commonly used was the Likert scale, but the range of the anchor points varied from 5 to 11 points. Other less frequently used scales were the visual analog scale (VAS 0–10 or 0–100) and a binary (Yes/No) response option.

**3.1.3 Study sample.** The selected studies with different datasets included a total of 4,393 participants, of which 3,678 (84%) were healthy participants and 715 (16%) clinical or at-risk populations. Mean age was 38.6 (SD = 12.0) years and 58% were female. Clinical or at-risk study populations were only included in 9 different datasets consisting of individuals with cardiovascular diseases (n = 135) [67], individuals at-risk for cardiovascular disease (e.g., pre- or mild hypertension) (n = 194) [38, 41], individuals diagnosed with post-traumatic stress

**Table 2. Associations with ambulatory blood pressure.**

| First author (year) | Sample n (females: males), mean age ± SD | Study length | Sampling frequency (sampling scheme) | Self-reported stress measures | Cardiovascular measures | Statistical analysis | Data points | Average completed prompts per subject (compliance %) | Results |
|---|---|---|---|---|---|---|---|---|---|
| Birk (2019)[a] | Healthy employees 373 (232:141), 52.0 ± np | 1 day | Every 30 minutes (fixed) | PS (past 10 min) | SBP (momentary) DBP (momentary) | MLM[1,2,3,5,7,11,13,14,18] | np | np | Positive association between PS and SBP and DBP |
| Thomas (2019)[b] Kamarck (2018)[b] | Healthy, working midlife adults 477 (241:236), 42.7 ± 7.3 | 4 days | Every 60 minutes (fixed) | AS (past 10 min) SS (past 10 min) | SBP (momentary) DBP (momentary) | MLM[11,12,13,14,16,17] | 25,386 | np | Positive association between AS and SBP and DBP and between SS and SBP; no association between SS and DBP |
| Cornelius (2018)[a] | Healthy employees 805 (482:323), 45.3 ± 10.3 | 1 day | Every 28–30 minutes (fixed) | SS (momentary) | SBP (momentary) DBP (momentary) | MLM[1,2,3,5,11,12,13,14,16,18] | 11,190[c] | 24.6 (np) | Positive association between SS and SBP; no association with DBP |
| Zawadzki (2016) | General community adults 39 (26:13), 51.697 ± 12.94 | 1 day | Every 20 minutes (fixed) | NA (momentary) | SBP (momentary) DBP (momentary) | MLM[1,2,8,12,13,14] | 1,368[c] | np | Positive association between SBP and NA; no association with DBP |
| Edmondson (2015)[a] | Healthy employees 858 (507:351), 45.2 ± 10.4 | 1 day | Every 28 minutes (fixed) | NA (momentary) | SBP (momentary) | MLM[1,2,5] | 20,916 | 24.4 (np) | Positive association between momentary NA and SBP |
| Lehman (2015) | Undergraduate students 68 (44:24), 20.6 ± 2.5 | 3 days | Every 42–78 minutes (random) | NA (momentary) SS (momentary) | SBP (momentary) DBP (momentary) | MLM[11,12,13,14] | 1,957 | 35.0 (np) | Marginal positive association between SS and NA (anxiety) and SBP; no association between SS or NA (anxiety) and DBP; no association between other NA items and SBP or DBP |
| Friedmann (2013) | Pet owners 32 (27:5), 60.5 ± 1.3 | 3 days | Every 20 minutes (fixed) | NA (momentary) | SBP (momentary) DBP (momentary) | GEE[12,18,19,20] | 2,430 | Median of 79.0 (np) | Negative association between NA and DBP; no association between NA and SBP |
| Lehman (2010) | Undergraduate students 99 (69:30), 21 ± np | 4 days | Every 60 minutes (fixed) | SS (past 10 min) | SBP (momentary) DBP (momentary) (every 60 min) | MLM[1,2,3,8,9,11,12,13,14,16] | 3,420 | 35.0 (np) | Positive association between SS and SBP and DBP |
| Ilies (2010) | University employees 67 (54:13), 42.6 ± 9.44 | 10 days | 4 per day, of which first 3 prompts randomly every 2 hours and the last prompt fixed at 4.45 p.m. (mixed) | NA (momentary) | SBP (momentary) DBP (momentary) | MLM[np] | 1,937 | np (72.3) | Positive association between NA and both SBP and DBP |
| Luecken (2009) | Undergraduate students from bereaved families 43 (26:17), 19.1 ± 1.4 Undergraduate students from non-bereaved families 48 (31:17), 20.0 ± 2.3 | 1 day | Every 30 minutes in 20-minute intervals (random) | ES (past hour) | SBP (momentary) DBP (momentary) | MLM[1,2,3,11,13,15,18] | 2,348 | 26.0 (81.0) | Positive association between ES and both SBP and DBP in combined sample |
| Pollard (2007) | Women with premenopausal women 26 (26:0), 39.0 ± 5.9 Women with postmenopausal 7 (7:0), 58.4 ± 4.4 | 2 days | 6 assessments per day (fixed) | PS (past hour) | SBP (momentary) DBP (momentary) | MLM[1,3,7,12,13,14] | 376 | np (94.9) | Positive association between PS and both SBP and DBP in combined sample |

(*Continued*)

**Table 2.** (*Continued*)

| First author (year) | Sample n (females: males), mean age ± SD | Study length | Sampling frequency (sampling scheme) | Self-reported stress measures | Cardiovascular measures | Statistical analysis | Data points | Average completed prompts per subject (compliance %) | Results |
|---|---|---|---|---|---|---|---|---|---|
| Meininger (2004) | Adolescents of 11–16 years 307 (np:np), np ± np | 1 day | Every 30 minutes (fixed) | PS (momentary) NA (momentary) | SBP (momentary) DBP (momentary) | MLM[1,2,4,5,11,12,18,19] | 8,428[c] | np (65.0) | No association between PS and SBP or DBP; negative association between NA (bored) and both SBP and DBP; marginal positive association between NA (angry) and SBP; no association between NA (irritable or sad) and SBP and DBP; no association between NA (angry) and DBP |
| Buckley (2004) | Vietnam combat veterans with PTSD 19 (0:19), 51.1 ± 3.3 Vietnam combat veterans without PTSD 17 (0:17), 53.4 ± 3.1 | 1 day | Every 20 minutes (fixed) | PS (momentary) | SBP (momentary) DBP (momentary) | MLM[3,4,7,5,8,11,12,13,14,15,18] | np | np | No association between PS and SBP or DBP in combined sample |
| Tsai (2003) | Women with normotensive 12 (12:0), 39.7 ± 7.7 | 1 day | Every 30 minutes between 6 a.m. and 10 p.m. and every 60 minutes between 10 p.m. and 6 a.m. (fixed) | PS (momentary) NA (momentary) | SBP (momentary) DBP (momentary) PP (momentary) | MLM[1,3,10,11,12,21] | 1,249[d] | np | Positive association between PS and both DBP and PP; no association between PS and SBP; no association between NA and DBP, SBP, DBP, or PP |
| Hawkley (2003) | Undergraduate students 70 (np:np), np ± np | 1 day | Every 45–120 minutes (random) | AS (momentary) | SBP (momentary) DBP (momentary) AP (momentary) | MLM[3,8,11,12,13,14,17] | 441 | np (90.0) | No association between AS and SBP, DBP, or MAP |
| Carels (2000) | High emotional responsive 81 (41:40), np ± np Low emotional responsive 81 (36:45), np ± np | 1 day | Approximately 4 times per hour between 7 a.m. and 11 p.m. and 2 times per hour between 11 p.m. and 7 a.m. (random) | NA (momentary) | SBP (momentary) DBP (momentary) | GEE[11,18,19] | 8,359[c] | 51.6 (np) | Positive association between NA and both SBP and DBP in combined sample |
| Picot (1999) | Afro-American female caregivers 37 (37:0), 55 ± 12.7 Afro-American female non-caregivers 38 (38:0), 50 ± 15.2 | 1 day | Every 30 minutes (fixed) | NA (momentary) | SBP (momentary) DBP (momentary) | GEE[1,3,4,6,7,8,13] | np | np | Negative association between NA (anger) and SBP in cargivers; marginal negative association between NA (anger) and DBP in caregivers; no association between NA (anger) and SBP or DBP in non-cargivers; no association with NA (unhappy) or NA (anxious) and SBP or DBP in either group |

(*Continued*)

**Table 2.** (Continued)

| First author (year) | Sample n (females: males), mean age ± SD | Study length | Sampling frequency (sampling scheme) | Self-reported stress measures | Cardiovascular measures | Statistical analysis | Data points | Average completed prompts per subject (compliance %) | Results |
|---|---|---|---|---|---|---|---|---|---|
| Kamarck (1998) | Full-time workers living with a partner 120 (64:56), 35 ± np | 6 days | Every 45 minutes (fixed) | NA (past 10 min) SS (past 10 min) | SBP (momentary) DBP (momentary) | MLM[11,12,13,14,16,17] | 13,080[c] | Np (99.0) | Positive association between NA and both SBP and DBP; no association between SS and SBP or DBP |

PTSD: posttraumatic stress disorder; AS: activity-related stress; SS: social stress; NA: negative affect; PS: perceived stress; ES: event-related stress; SBP: systolic blood pressure; DBP: diastolic blood pressure; PP: pulse pressure; MAP: mean arterial pressure; MLM: multilevel modelling; GEE: generalized estimating equation; np: not provided; marginal association means $p < .1$.

Studies controlled for the effects of relevant time-invariant factors:

[1]age,

[2]sex,

[3]body shape factors (body-mass index/ waist-hip ratio),

[4]socioeconomic factors (education/ income/ employment status/ socioeconomic status, mother's education),

[5]racial or minority-related factors (race, ethnicity, minority status),

[6]medication status,

[7]smoking status,

[8]health-related risk factors (general health/ psychopathology/ family history of hypertension/ cardiovascular risk factors/ hypertension diagnosis),

[9]sleep,

[10]menstrual cycle; and time-variant factors:

[11]posture,

[12]physical activity,

[13]substance use (intake of alcohol, nicotine, caffeine, or recreational drugs),

[14]food intake,

[15]medication use,

[16]temperature,

[17]talking,

[18]location,

[19]mood,

[20]presence of another person,

[21]time.

+: significant positive association; ~+: marginally significant positive association; ~-: marginally significant negative association; -: significant negative association; ·: no association.

[a] These studies report on the same sample with different analyses.

[b] These studies report on the same sample with similar analyses, we used the results from Thomas (2019) for all our purposes.

[c] These numbers are estimated based on average number of data entries.

[d] This number was inferred from the degrees of freedom reported.

disorder (n = 118) [37, 56], individuals diagnosed with borderline personality disorder (n = 50) [59], individuals diagnosed with psychosis (n = 20) [62], individuals diagnosed with psychosis and individuals at-risk for psychosis (n = 67) [63], individuals at-risk for psychopathology (n = 91) [48], and individuals diagnosed with substance use disorder (n = 40) [61]. More information on sample characteristics is presented in the S2 File.

**3.1.4 Study methods and compliance.** For self-reported daily life stress, the most frequently used diary design had a random sampling scheme (50% of studies) or a fixed sampling scheme (47% of studies). One study (3%) used a mixed sampling of both random and fixed

**Table 3. Associations with heart rate.**

| First author (year) | Sample n (females: males), mean age ± SD | Study length | Sampling frequency (sampling scheme) | Self-reported stress measures (timing relative to beep) | Cardiovascular measures (timing relative to prompts) | Statistical analysis | Data points | Average completed prompts (compliance %) | Results |
|---|---|---|---|---|---|---|---|---|---|
| Cornelius (2018) | Healthy employees 805 (482:323), 45.3 ± 10.3 | 1 day | Every 28–30 minutes (fixed) | SS (momentary) | HR (momentary) | MLM[1, 2, 3, 5, 11, 12, 13, 14, 16, 18] | 11,190[c] | 24.6 (np) | No association between SS and HR |
| Dennis (2018)[a] | Young adults with trauma 178 (100:78), 28.8 ± 5.54 | 1 day | Every 2–3 hours (np) | NA (momentary) | HR (next 5 min) | MLM[1, 2, 3, 4, 7, 11, 12] | 1,221.08[c] | 6.9 (85.0) | No association between NA and HR |
| Krkovic (2018) | Individuals with psychotic-like experiences 67 (48:19), 23.1 ± 4.6 | 1 day | Every 20 minutes between 9 a.m. and 10 p.m. (fixed) | PS/NA (past 20 min) | HR (past 20 min) | Correlation analysis at the within-subject level | 2,042 | np (81.0) | No association between PS/NA and HR |
| Dennis (2017)[a] | Young adults with trauma 197 (100:97), 28.87 ± 5.57 | 1 day | Every 2–3 hours (random) | NA (past 5 min) | HR (next 5 min) | MLM[1, 2, 3, 12] | 1,369.15[c] | 7.0 (np) | No association between NA and HR |
| Dennis (2016)[a] | Individuals with PTSD 99 (49:50), 30.3 ± 5.4 Individuals with no-PTSD 120 (64:56), 27.8 ± 5.47 | 1 day | Every 2–3 hours (random) | PS (momentary) | HR (next 5 min) | MLM[1, 7, 11, 12] | 1,620.6[c] | 7.4 (np) | No association between PS and HR in combined sample |
| Johnston (2016) | Nurses 100 (93:7), 36.4 ± 9.9 | 2 days | 8 per day (np) Every 90 minutes in 30-minute intervals (random) | AS (past 10 min) | HR (past 10 min) | MLM[12] | 1,453 | np (98.5) | No association between AS and HR |
| Lehman (2015) | Undergraduate students 68 (44:24), 20.6 ± 2.5 | 3 days | Every 42–78 minutes (random) | NA (momentary) SS (momentary) | HR (momentary) | MLM[11, 12, 13, 14] | 1,957 | 35.0 (np) | Marginal positive association between SS or NA (anxiety) and HR; no association between NA (shame), NA (embarrassment), or NA (anger) and HR |
| Kennedy (2015) | Polydrug users 40 (10:30), 41.4 ± 8.3 | 23 days | 3 per day (random) | PS (momentary) SS (momentary) | HR (past and next 15 min) | MLM[12, 19] | 2,329 | np | Positive association between PS and HR and between SS and HR |
| Riediger (2014) | Healthy individuals 92 (51:41), 42.4 ± 19.0 | 2 days | 6 per day every 2 hours (random) | PS (momentary) | HR (momentary) | MLM[1, 12] | 644[c] | 7.0 (58.3) | Positive association between PS and HR; no association between PS (squared) and HR |
| Lehman (2010) | Undergraduate students 99 (69:30), 21 ± np | 4 days | Every 60 minutes (fixed) | SS (past 10 min) | HR (momentary) | MLM[1, 2, 3, 8, 9, 11, 12, 13, 14, 16] | 3,420 | 35 (np) | No association between SS and HR |
| Ilies (2010) | University employees 67 (54:13), 42.6 ± 9.44 | 10 days | 4 per day, of which first 3 prompts randomly every 2 hours and the last prompt fixed at 4.45 p.m. (mixed) | NA (momentary) | HR (momentary) | MLM[not reported] | 1,937 | np (72.3) | Positive association between NA and HR |

*(Continued)*

**Table 3.** (*Continued*)

| First author (year) | Sample n (females: males), mean age ± SD | Study length | Sampling frequency (sampling scheme) | Self-reported stress measures (timing relative to beep) | Cardiovascular measures (timing relative to prompts) | Statistical analysis | Data points | Average completed prompts (compliance %) | Results |
|---|---|---|---|---|---|---|---|---|---|
| Kimhy (2010) | Individuals with psychosis 20 (10:10), 30.6 ± 8.4 | 2 days | 10 per day between 10 a.m. to 10 p.m. (random) | NA (momentary) (every 1 to 143 min) | HR (past and next 5 minutes) | MLM[6] | 300 | np (79.0) | No association between NA and HR |
| Pieper (2010)[b] | Teachers 73 (24:49), 46.7 ± 9.5 | 4 days | Between 8 a.m. to 10 p.m. in 45–75 minute intervals (random) | PS (momentary) NA (momentary) ES (past hour) | HR (15 minutes interval Following retrospectively reported event) | MLM[1, 2, 3, 11, 12, 13, 19, 21] | 1,957 | 26.8 (np) | Marginal positive association between PS and HR; no association between NA and HR; no association between ES and HR |
| Ebner-Priemer (2008) | Individuals with BPD 50 (50:0), 31.3 ± 8.1 Healthy individuals 50 (50:0), 27.7 ± 6.8 | 1 day | Every 10–20 minutes (random) | PS (momentary) | Additive HR (10–20 minutes prior to prompt) | MLM[7, 12, 21] | 5,410 | 52.3 (bpd) 55.9 (hp) (np) | Positive association between PS and HR in combined sample |
| Pieper (2007)[b] | Teachers 73 (24:49) 46.7 ± 9.5 | 4 days | 14 per day between 8 a.m. to 10 p.m. (random) | ES (past hour) | HR (past 5–60 minute, on average 6.85 minutes) | MLM[1, 2, 3, 8, 13, 19, 21] | 2,653 | 36.3 (64.8) | Positive association between ES and HR |
| Pollard (2007) | Women with premenopausal 26 (26:0), 39.0 ± 5.9 Women with postmenopausal 7 (7:0), 58.4 ± 4.4 | 2 days | 6 per day (fixed) | PS (past hour) | HR (momentary) | MLM[1, 3, 7, 12, 13, 14] | 376 | np (94.9) | Positive association between PS and HR in combined sample |
| Buckley (2004) | Vietnam combat veterans with PTSD 19 (0:19), 51.1 ± 3.3 Vietnam combat veterans without PTSD 17 (0:17), 53.4 ± 3.1 | 1 day | Every 20 minutes (fixed) | PS (momentary) | HR (momentary) | MLM[3, 4, 5, 7, 8, 11, 12, 13, 14, 15, 18] | np | np | No association between PS and HR in combined sample |
| Hawkley (2003) | Undergraduate students 70 (np:np), np ± np | 1 day | Every 45–120 minutes (random) | AS (momentary) (every 45–120 min) | HR(momentary) (every 45–120 min) | MLM[3, 8, 11, 12, 13, 14, 17] | 441 | np (90.0) | No association between AS and HR |
| Tsai (2003) | Women with normotensive 12 (12:0), 39.7 ± 7.7 | 1 day | Every 30 minutes from 6 a.m. to 10 p.m. and every 60 minutes between 10 p.m. and 6 a.m. (fixed) | PS (momentary) NA (momentary) | HR (momentary) | MLM[1, 3, 10, 11, 12, 21] | 1,249[d] | np | Positive association between NA and HR; no association between PS and HR |
| Carels (2000) | High emotional responsive 81 (41:40), np ± np Low emotional responsive 81 (36:45), np ± np | 1 day | 4 times per hour between 7 a.m.–11 p.m. and 2 times per hour between 11 p.m.–7 a.m. (random) | NA (momentary) | HR (momentary) | GEE[11, 18, 19] | 8,359.2[c] | 51.6 (np) Number of waking hour prompts not reported | Positive association between NA and HR in combined sample |

(*Continued*)

**Table 3.** (Continued)

| First author (year) | Sample n (females: males), mean age ± SD | Study length | Sampling frequency (sampling scheme) | Self-reported stress measures (timing relative to beep) | Cardiovascular measures (timing relative to prompts) | Statistical analysis | Data points | Average completed prompts (compliance %) | Results |
|---|---|---|---|---|---|---|---|---|---|
| Kamarck (1998) | Fulltime workers living with a partner 120 (64:56), 35 ± np | 6 days | Every 45 minutes (fixed) | NA (momentary) AS (momentary) SS (momentary) | HR (momentary) (every 45 min) | MLM[11, 12, 13, 14, 16, 17] | 13,080[c] | np (99.0) | No association between NA and HR; positive association between AS and HR; no association between SS and HR |

PTSD: posttraumatic stress disorder; BPD: borderline personality disorder; HP: healthy participants; SS: social stress; NA: negative affect; PS: perceived stress; AS: activity-related stress; ES: event-related stress; HR: heart rate; MLM: multilevel modelling; GEE: generalized estimating equation; np: not provided.

Studies controlled for the effects of relevant time-invariant factors:

[1]age,

[2]sex,

[3]body shape factors (body-mass index/ waist-hip ratio),

[4]socioeconomic factors (education/ income/ employment status/ socioeconomic status, mother's education),

[5]racial or minority-related factors (race, ethnicity, minority status),

[6]medication status,

[7]smoking status,

[8]health-related risk factors (general health/ psychopathology/ family history of hypertension/ cardiovascular risk factors/ hypertension diagnosis),

[9]sleep,

[10]menstrual cycle; and time-variant factors:

[11]posture,

[12]physical activity,

[13]substance use (intake of alcohol, nicotine, caffeine, or recreational drugs),

[14]food intake,

[15]medication use,

[16]temperature,

[17]talking,

[18]location,

[19]mood,

[20]presence of another person,

[21]time.

+: significant positive association; ~+: marginally significant positive association; ~-: marginally significant negative association; -: significant negative association; ·: no association; np: not provided; marginal association means p < .1.

[a] These studies report on the same sample with different analyses.

[b] These studies report on the same sample with different analyses.

[c] These numbers are estimated based on average number of data entries.

[d] This number was inferred from the degrees of freedom reported.

sampling schemes [43]. The random sampling schemes varied from 2 times per hour to 6 times per day and the fixed sampling scheme varied from every 20 minutes to three times per day. Studies measuring blood pressure (N = 19) were mostly 1-day study protocols in (N = 11; 56%) studies with an average of 2.5 (SD = 2.4; range = 1–10) study days. The frequency of blood pressure assessments varied from every 20 minutes to 5 times per day. For heart rate studies (N = 21), the length of the study days was on average 3.0 (SD = 5.0; range = 1–23) days, with the most common protocol being also a 1-day study protocol (N = 10; 48%). Study length in heart rate variability studies (N = 12) was also heterogeneous ranging from 1 to 6 days with

**Table 4. Associations with heart rate variability.**

| First author (year) | Sample n (females:males), mean age ± SD | Study length | Sampling frequency (sampling scheme) | Self-reported stress measures | Cardiovascular measures | Statistical analysis | Data points | Average completed prompts per subject (compliance %) | Results |
|---|---|---|---|---|---|---|---|---|---|
| Schilling (2020) | Police officers 201 (72:129), 38.6 ± 10.1 | 2 days | Once per hour between 12 a.m. and 7 p.m. for all shift workers and between 9 a.m. and 5 p.m. for regular office workers (random) | PS (momentary) | RMSSD (10 minutes following prompt) | MLM[12,19] | np | 6.7 (80.9) | No association between PS; no association between NA and RMSSD |
| Schmid (2020) | Teachers 101 (70:31), 42.9 ± 11.5 | 2 days | Every 1–2 hours between 7.30 a.m. and 9 p.m. (random) | AS (momentary) | lnRMSSD (5 minutes following prompt) | MLM[1,3,21] | 669 | 10.2 (86.0) | No association between AS and lnRMSSD |
| Schwerdtfeger (2019) | Firefighters 43 (0:43), 32.7 ± 6.9 | 1 day | Every 60 minutes (random) | NA (momentary) | lnRMSSD (6 minutes following prompt) | MLM[1,7,12,21] | 623 | 5.2 (np) | No association between NA and lnRMSSD |
| Dennis (2018)[a] | Young adults with trauma 178 (100:78), 28.8 ± 5.54 | 1 day | Every 2–3 hours (random) | NA (momentary) | LF-HRV HF-HRV (5 minutes following prompt) | MLM[1,2,3,4,7,11,12] | 1,221[c] | 6.9 (85.0) | No association between NA and LF-HRV or HF-HRV |
| Dennis (2016)[a] | Patients with PTSD 99 (49:50), 30.3 ± 5.4 no-PTSD patients 120 (64:56), 27.8 ± 5.47 | 1 day | Every 2–3 hours (random) | PS (momentary) | LF-HRV (bpm) HF-HRV (5 minutes following prompt) | MLM[1,7,11,12] | 1,621[c] | 7.4 (np) | Negative association between PS and LF-HRV; marginal negative association between perceived stress and HF-HRV in combined sample |
| Schwerdtfeger (2014) | Healthy adults 117 (67:50), 27.8 ± 5.4 | 3 days | Every 50–80 minutes between 8 a.m. to 10 p.m. (random) | NA (momentary) | lnRMSSD (past 5) | MLM[1,2,3,12,13,18,19] | 3,346 | np (81.0) | No association between NA and RMSSD |
| Kimhy (2010) | Patients with psychosis 20 (10:10), 30.6 ± 8.4 | 2 days | 10 per day between 10 a.m. to 10 p.m. (random) | NA (momentary) (every 1–143 min) | LF-HRV HF-HRV (5 minutes prior to and following prompt) | MLM[6] | 300 | np (79.0) | Negative association between NA and HF-HRV; no association between NA and LF-HRV |
| Pieper (2010)[b] | Teachers 73 (24:49), 46.7 ± 9.5 | 4 days | Between 8 a.m. to 10 p.m. in 45–75 minute intervals (random) | PS (momentary) NA (momentary) ES (past hour) | lnMSSD (15 minutes prior to diary) | MLM[1,2,3,8,12,13,19,21] | 1,957 | 26.8 (np) | No association between PS, NA, or ES and lnMSSD |
| Ebner-Priemer (2008) | BPD patients 50 (50:0), 31.3 ± 8.1 Hhealthy controls 50 (50:0), 27.7 ± 6.8 | 1 day | Every 10–20 minutes (random) | PS (momentary) | Additive HF-HRV (10–20 minutes prior to prompt) | MLM[6,12,21] | 5,410 | 52.3 (bpd) 55.9 (hp) (np) | No association between PS and additive HF-HRV in combined sample |
| Pieper (2007)[b] | Teachers 73 (24:49) 46.7 ± 9.5 | 4 days | 14 per day between 8 a.m. to 10 p.m. (random) | ES (past hour) | lnRMSSD (5–60 minute interval in hour prior to prompt, on average 6.85 minutes) | MLM[1,2,3,8,13,19,21] | 2,653 | 36.3 (64.8) | Marginal negative association between event stress and lnRMSSD |

(*Continued*)

**Table 4.** (Continued)

| First author (year) | Sample n (females:males), mean age ± SD | Study length | Sampling frequency (sampling scheme) | Self-reported stress measures | Cardiovascular measures | Statistical analysis | Data points | Average completed prompts per subject (compliance %) | Results |
|---|---|---|---|---|---|---|---|---|---|
| Bacon (2004) | Coronary artery disease patients 135 (41:94), 63 ± 10.0 | 2 days day not reported | Every 20-minutes (fixed) | PS (momentary) NA (momentary) | Momentary lnHF-HRV momentary lnLF-HRV (1 minute during diary) | MLM[1,6,11] | 15,390[c] | 114.0 (np) | Negative association between PS and both HF-HRV and LF-HRV, and between NA and both HF-HRV and LF-HRV |
| Sloan (1994) | Healthy individuals 33 (2:31), 37.9 ± 12.8 | 1 day | Once per hour (random) | NA (momentary) | Mean rr interval lnLF-HRV lnHF-HRV (5 minutes prior to diary) | MLM[11] | 362 | np | Negative association between NA and mean RR interval; no association between NA and LF-HRV or HF-HRV |

PTSD: post-traumatic stress disorder; BPD: borderline personality disorder; AS: activity-related stress; NA: negative affect; PS: perceived stress; ES: event-related stress; LF-HRV: low-frequency heart rate variability; HF-HRV: high-frequency heart rate variability; RMSSD: root mean square of successive differences; MSSD: mean square of successive differences; MLM: multilevel modelling; np: not provided; marginal association means p < .1.

Studies controlled for the effects of relevant time-invariant factors:

[1]age,

[2]sex,

[3]body shape factors (body-mass index/ waist-hip ratio),

[4]socioeconomic factors (education/ income/ employment status/ socioeconomic status, mother's education),

[5]racial or minority-related factors (race, ethnicity, minority status),

[6]medication status,

[7]smoking status,

[8]health-related risk factors (general health/ psychopathology/ family history of hypertension/ cardiovascular risk factors/ hypertension diagnosis),

[9]sleep,

[10]menstrual cycle; and time-variant factors:

[11]posture,

[12]physical activity,

[13]substance use (intake of alcohol, nicotine, caffeine, or recreational drugs),

[14]food intake,

[15]medication use,

[16]temperature,

[17]talking,

[18]location,

[19]mood,

[20]presence of another person,

[21]time.

+: significant positive association; ~+: marginally significant positive association; ~-: marginally significant negative association; -: significant negative association; ·: no association.

[a] These studies report on the same sample with different analyses.

[b] These studies report on the same sample with different analyses.

[c] These numbers are estimated based on average number of data entries.

an average of 2.1 (SD = 1.1, range = 1–4) study days, with the most common protocol being again a 1-day (N = 4; 33%) or a 2-day (N = 4; 33%) study protocol. The frequency of HR/HRV assessments varied from 15-seconds intervals to only 5 times per day.

The data collection method for self-reported daily life stress was reported in 27 (75%) studies, of which the most commonly used device was a dedicated device (i.e., built-in software on a personal digital assistant, mobile phone software, or handheld computer) in 17 (47%) studies. Six (6%) studies used a smartphone application and four (10%) studies used a traditional paper-and-pencil diary. Ambulatory devices to detect daily life stress varied across the studies (S1 Table). Ten different devices were used to measure blood pressure, with the most common device being the Spacelabs model 90207 (N = 5; 31% for SBP and 22% for DBP) [39, 40, 42, 50, 53]. Fifteen different devices were used to measure heart rate, with the most common devices being a Holter device in four (19%) studies [56, 58, 62, 74], Spacelabs model 90207 in three studies [39, 42, 53], and VU-AMS in two (10%) studies [64, 65]. For heart rate variability, seven different devices were identified and the most used devices were either a Holter monitor [56, 58, 62] or a Movisens EcqMove [70–72] device in three studies each (25%).

The overall compliance rate for self-reported stress assessments was reported only in 15 (42%) studies, ranging from 58% to 99% with an average of 81% (SD = 12.1) compliance rate. Outliers, artifacts, or other forms of thresholds used to minimize the noise in the retrieved objective data for cardiovascular stress measures were reported in 29 (81%) studies. Only 11 of these studies (31% of all included studies) reported the values of missing data due to outliers, artifacts, or other thresholds in cardiovascular measures.

**3.1.5 Methodological quality of the included studies.** The overall methodological quality of the studies was fair (S2 Table). Most of the studies were characterized by good study reporting (domains 1 to 10). The major issue in the quality of the studies was the lack of external validity related to insufficient reporting of the proportion of the source population where the participants were derived. Also, the quality assessment revealed an increased risk of selection bias in recruitment over a period of time, which was not clearly reported in most of the studies.

## 3.2 Descriptive results on the associations between self-reported stress and cardiovascular measures

The 36 studies yielded a total of 135 separate analyses associating self-reported stress measures with ambulatory cardiovascular measures. Overall, studies reporting 38 out of 135 analyses (28%) revealed statistically significant associations in the expected direction. Another nine (7%) analyses were only marginally significant (i.e., p < .10) in the expected direction. Included studies in this review used different statistical approaches to test associations between self-reported stress and cardiovascular measures; 32 studies used multilevel modelling (MLM), three studies used generalized estimating equation (GEE), and one study used correlation analysis at the within-subject level.

**3.2.1 Self-reported stress.** Of the 27 analyses on perceived stress, 13 (48%) showed significant associations in the expected direction with ambulatory cardiovascular measures and 2 (7%) were marginally significant. For NA, this was 15 (22%) out of 69 analyses showing a significant association in the expected direction and three (4%) a marginally significant association in the expected direction. When only considering analyses on all-high-arousal scales (25 analyses), eight (32%) of the analyses were significant in the expected direction, two (8%) were marginally significant in the expected direction. Regarding situational stress measures, 10 (26%) out of 38 analyses yielded significant results in the expected direction and four (10%) analyses were marginally significant (activity-related stress: 2/17 [12%] significant, 1/17 [6%]

marginally significant; social stress: 5/15 [33%] significant, 2/15 [13%] marginally significant; event-related stress: 3/6 [50%] significant, 1/6 [17%] marginally significant).

**3.2.2 Cardiovascular measures.** The 18 studies reporting on associations with BP made use of 16 different datasets and provided a total of 76 analyses. Of those analyses, 22 (29%) indicated a significant positive association with self-reported stress and three more (4%) showed a marginally significant positive association. For SBP, 12 out of 36 analyses (33%) showed a significant association in the expected direction with another three (8%) only reaching marginal significance; 10 out of 36 (28%) analyses were significant in the expected direction for DBP. PP was positively associated with self-reported stress in one analysis and unrelated in the other (50%), for MAP no (marginally) significant associations were reported in either of the two analyses.

As with BP, analyses of associations with self-reported stress for HR provided significant associations only in a minority of analyses. Results in the expected direction were obtained in 9 (26%) out of 35 analyses. Marginally significant positive associations were found in four (11%) analyses.

Seven (29%) out of 24 analyses on associations with HRV showed significant associations in the expected direction; two (8%) more only reached marginal significance. Looking at only the 13 analyses on frequency-domain measures (i.e., HF/LF-HRV), the distribution showed a significant association in the expected direction in six (46%) of the analyses and a marginally significant association in the expected direction in one (8%) of the analyses. For the eleven analyses on time-domain measures (i.e., mean r-r interval, MSSD, RMSSD), this was one (9%) and one (9%) out of 11 analyses, respectively.

**3.2.3 Study populations.** In healthy participants, 24 (24%) out of 100 analyses were significant in the expected direction and 8 (8%) were marginally significant. In contrast, of the 22 analyses done in patient samples (or a combined sample including patients), nine (41%) analyses showed a significant and one (5%) analysis a marginally significant association in the expected direction. At-risk samples showed a significant association in the expected direction in five (71%) out of seven analyses.

**3.2.4 Study methods and compliance.** Associations per study characteristic (i.e., length of the study and sampling [fixed vs. random] technique) and used devices are reported in the S1 Table. Due to the lack of reporting compliance in 21 (58%) of the studies and large heterogeneity in reporting the frequency of the assessments (Tables 2–4), no descriptive analyses were conducted of their possible influences on the associations. Based on our descriptive analyses of the extracted data available, studies that found association with HR included more study days (average of 6.0 study days) than studies that did not find associations with HR (average of 2.2 study days). Also, there was a marginally significant association between self-reported stress and cardiovascular analyses in study protocols using a fixed sampling compared to a random sampling scheme (S1 Table).

Regarding the used devices, we found that within the analyses using Spacelabs 90207 equipment, significant associations were reported in 15% for SBP (2/13), 18% for DBP (3/17), and 17% for HR (1/6) of the analyses. For the Spacelabs 90217 equipment, significant associations were found in 25% for SBP (2/8) and 25% for DBP (2/8) of the analyses. For HRV, the most commonly used device (i.e., Holter monitoring) was used in six analyses resulting in significant association only in 33% (2/6) of the analyses.

## 4 Discussion

The purpose of this review was to investigate how self-reported stress and ambulatory cardiovascular measures are operationalized in daily-life studies, and what the evidence is for an

association between self-reported stress and cardiovascular responses indicative of ANS activity in these studies. Based on our descriptive synthesis, much heterogeneity was evident between studies in terms of self-reported stress assessment, methodology, devices, and study population. Overall, the studies reviewed here showed an association in the expected direction between self-reported stress and cardiovascular parameters in 28% of analyses (35% when including marginally significant associations). This percentage is slightly higher than the 25% found in a previous systematic review of 12 laboratory studies investigating associations between self-reported and cardiovascular measures of stress [24]. Results did, however, show variability among self-reported stress and cardiovascular measures. Significant and marginally significant associations were observed in analyses on perceived stress measures (55%), and least likely (18%) in analyses on activity-related stress measures. With (marginally) significant associations in 54% of the analyses, frequency domain measures of HRV yielded the most positive results, whereas MAP and SCL were not associated with self-reported stress.

## 4.1 Descriptive findings of the association between self-reported stress and cardiovascular measures in daily life

The fact that self-reported stress and cardiovascular measures were only significantly associated in less than a third of all studies is not completely unexpected. For instance, the experienced intensity of an emotion is considerably stronger associated with behavior than with physiology, emphasizing the social nature of this inter-system coherence [28]. Interestingly, Mauss et al. [28] found that the association between perceived intensity and both behavior and physiology was weaker for negative emotions than for positive emotions. According to them, this may have to do with social norms according to which individuals are expected to control negative emotions, more so than positive emotions, creating incoherence. This may certainly explain why stress, while not technically an emotion, does often not present itself as a very cohesive construct across different systems. Still, it must be emphasized that neither based on prior research, nor on the current review, can we conclude that there is no association between these systems. What we can conclude is that the likelihood of finding an association between a single self-report stress measure and a single physiological variable is moderate at best. On the other hand, machine-learning models, such as support-vector machines, random forest models, and Bayesian networks, have been able to, based on a set of physiological features, predict self-reported stress with high accuracy [75–78]. Such findings encourage the idea that there is cohesion among these systems, even though "simpler" models may not be able to detect its complex patterns. Moreover, there is much inter-individual variation in the physiological stress response, calling for personalized models that can capture the individual's physiological signature. In the end, a single stress measure provides incomplete information about an individual's stress level, and in order to provide a better picture, stress should be assessed on all three levels: experientially, behaviorally, and physiologically, while keeping in mind its highly personalized character.

## 4.2 Self-reported stress assessment

The large heterogeneity in self-reported stress measures used in the studies reviewed here complicates any inference to be made. The most homogeneous self-reported stress measure types, perceived stress and event-related stress, showed the most consistent results. Taking a liberal stance, of all analyses reported here on perceived stress in daily life, more than half showed a significant or a marginally significant association with cardiovascular variables in the expected direction. Given that NA scales that only included high-arousal items more often showed associations with cardiovascular measures than scales including low-arousal items, it can be argued

that they better capture the subjective experience of physiological arousal, similar to the perceived stress measure. Inclusion of such low arousal items may obscure a potential association between the high-arousal items and physiology, hence should be avoided.

Results for event-, activity-, and social-related stress showed associations in only 26% of the analyses. A factor of influence here could be the variability between self-reported stress items of the same type. Social stress in particular varied widely in its operationalization, with only two studies having a similar approach. Although this heterogeneity may explain part of the mixed findings, it hampers the comparability of the studies. For activity-related stress, although operationalizations differed between and within studies, most measures were variations to the demand and control model of Karasek [20], meaning they are rooted in theory. However, the 12% success rate should, at the least, urge researchers to re-evaluate self-reported activity-related stress measures in association with cardiovascular measures. Event-related stress, on the other hand, is a retrospective measure, meaning that associations with physiology assessed up to an hour later, or averaged over the past hour, may be generally weaker. Indeed, the time-frame in which the stressful events were allowed to occur seems to play a role, with better results for shorter (i.e. 30 minutes) than longer (i.e. 45–75 minutes) time-windows. However, based on only a handful of studies, these conclusions need to be taken with caution. Still, out of the three types of situational stressors, event-related stress performed best, with (marginally) significant associations in two-thirds of all analyses.

Taken together, we observed high heterogeneity in between and within studies, which also indicates that there seems to be no consensus on how to assess ambulatory self-reported stress in the current research field of detecting stress in daily life. Factors that are likely contributing to this heterogeneity are decisions made on study sample, measures, sampling methods, and statistical analyses, in addition to the variability within the measures themselves. Future studies should opt for more evidence-based measures. Based on this review, perceived stress, high-arousal NA, and event-related stress measures with short time intervals were most convincing, with associations in the expected direction in about 40–60% of all analyses. Needless to say that this is far from convincing evidence in favor of an association between self-reported stress and cardiovascular measures in daily life.

## 4.3 Study populations

This descriptive review's findings suggest that studies investigating the associations between self-reported stress and cardiovascular measures in daily life are relatively more often reported in patient samples than in samples of healthy volunteers. In most cases, patient samples consisted of individuals diagnosed with a psychiatric disorder (i.e. PTSD, psychosis, BPD, substance abuse); in one study, they were diagnosed with cardiovascular disease. Although these conditions are very different, stress has been shown to play a role in all of them. Moreover, analyses on individuals at risk for stress-related disorders showed significant associations in the expected direction in five out of seven analyses. In psychosis, even compared to patients, at-risk individuals show an increased affective reactivity to daily-life stress [11, 79]. Possibly, increased reactivity denotes a stronger coupling of subjective experience and physiology, which results in stronger associations between the stress systems.

One possible explanation for sample differences is that patients and individuals at risk may experience more fluctuations in their stress levels during the day, providing for more variability in the data and hence increase the likelihood of finding an association. An alternative explanation comes from the fact that, for all studies reviewed here, the onset of stress most likely occurred at some point in between diary entries. Consequently, all results reported here reflect associations in the recovery phase of acute stress or even chronic stress levels. The recovery phase is an

interesting, yet often overlooked phase of the stress response that is delayed in individuals at risk for and at early stages of mental illness [79, 80]. Interindividual differences in recovery may therefore contribute to a weakening of the association over time. This could explain the finding that associations between self-reported stress and cardiovascular measures were found in the majority of studies on individuals with a clinical diagnosis or individuals at risk, as recovery may be delayed in these populations. For healthy participants, an association between the two measures may be present during the acute stress response but diminished by the time of the first assessment moment. This possibility is, however, not supported by the findings of Campbell and Ehlert [24] reporting associations to acute stress in only about 25% of all studies.

## 4.4 Study methods and compliance

Our descriptive analyses identified a vast heterogeneity in used devices. High heterogeneity in used devices may be explained by the increase in technological developments that enables more ambulatory devices to be used in a real-world setting. However, it also indicates that one device is not shown to be more beneficial to observe associations between self-reported stress and cardiovascular measures, although our findings are limited to show this direction statistically. We also acknowledged that the validation of these used devices was not always well reported, and some devices have shown controversial results on its validity. For example, the most commonly used ABP monitor (Spacelabs model 90207) has raised concerns of its possible limitation as the readings were altered by venous blood redistribution [81], and a direct effect of cuff inflation lead to the underestimation of ongoing HR during a cuff-based ABP [82]. Besides these limitations, the model has been shown to be a valid monitoring tool to measure ABP [82, 83].

For study protocols, our findings did not indicate differences between the associations based on the study length except for studies that used HR devices. Based on the data available within HR analyses, our findings indicated that associations were more often found in studies with longer study period (i.e., average of 6.0 study days) than studies with less study days (average of 2.3 study days). This finding may suggest that more study days may be recommended for studies combining self-reported stress and HR measure. However, this needs to be interpreted with caution while only nine (26%) of the analyses found significant association from the total HR analyses and the length of the study is also driven by the research questions and hypotheses, and therefore, it is challenging to recommend a certain type of protocol to be more favorable of another. Furthermore, our findings showed differences in used sampling techniques. As can be expected, studies using blood pressure measures more often opted for a fixed sampling scheme. Compared to studies with random sampling schemes, those using a fixed sampling scheme tended to show better associations between self-reported stress and cardiovascular measures (S1 Table). However, these findings need to be interpreted carefully as the distribution of sampling techniques varied between and within studies.

From a study quality perspective, most studies reported the description of the study procedure in detail, but compliance rates of the diary protocol were only reported in half of the studies. Also, thresholds to be used to minimize the noise in physiological data were fairly consistently reported, but the actual amount of excluded data due to this was only reported in a few studies. These methodological findings call for a more precise approach when it comes to the description of the missing data to increase the quality of their study protocol. Lastly, one noteworthy methodological finding was that the scales and the chosen items to measure self-reported stress were heterogeneous across the studies. This confirms that, although numbers are increasing, studies on daily life stress are still scarce, and the definitions of stress are driven by different approaches of individual studies.

### 4.5 Strengths and limitations

One strength of this systematic review is that it provides the first literature overview of associations between self-reported stress and cardiovascular measures measured simultaneously in a real-world setting. Also, this review provides insight into the relationship between self-reported stress and cardiovascular measures in a real-world setting. At the same time, we need to consider some limitations. Firstly, the studies included in this review were very heterogeneous in terms of their statistical analyses. Although most studies used MLM, the large diversity between models prevented us from conducting meta-analyses based on the reported values of the associations per stress measure. Therefore, an accurate estimate of the strength of the association could not be provided. Future investigations should put in effort to mirror the statistical models and methods previously used to estimate the association between self-reported and cardiovascular measures of stress so that meta-analyses can be conducted. Secondly, all our conclusions are based on group analyses. There are large inter-individual differences in stress responses and these findings have to be interpreted bearing that in mind. Thirdly, the overall level of the study quality was fair, where the major lack of reporting was related to external validity and selection bias (S2 Table). The most concerning issues in study quality were in reporting the proportion of the population source and the time period of the recruitment. Therefore, more attention should be given to the reporting of the population source and the timing period of the recruitment in future studies. Finally, we chose the somewhat arbitrary cut-off of 1-day monitoring (i.e., 24h) as an exclusion criterion. ESM is a method that aims to capture multiple snapshots of an individual's everyday life, and study periods of less than a day may not be long enough to do so. Despite these limitations, we believe that our review gives important insights into the use of self-reported stress and cardiovascular measures in a real-world setting, which hopefully will raise the awareness to investigate this topic more in the future.

## 5 Conclusions and recommendations

Overall, this systematic review shows that daily-life self-reports of stress and cardiovascular measures were associated in 28% of analyses (35% when including marginally significant findings). Analyses on perceived stress, high-arousal NA, or event-related stress measures, frequency-domain HRV measures, or in patients or at-risk populations had a larger proportion of analyses that were statistically significant; analyses on activity-related stress or low-arousal NA measures, time-domain HRV measures, or studies using Spacelabs ambulatory blood pressure equipment had lower rates. Therefore, based on this review, we recommend researchers to use the following when investigating the association between self-reported stress and cardiovascular measures:

- Perceived stress or high-arousal NA self-report measures

- High-quality wearable sensors that have been validated in ambulatory settings

- Time windows of max 30 minutes prior to self-report for the calculation of continuously assessed cardiovascular measures

- Study periods of at least 6 days with multiple assessments per day to capture enough variability in both measures

- Multilevel modelling for the statistical analyses

Although the results reviewed here are far from convincing, even when accounting for a possible publication bias, if the experiential stress response would be unrelated to its

physiological counterpart, it would be a strong claim that all analyses showing a significant association in the expected direction rely solely on type-I errors. As stress is marked by an increase in perceived stress and activation of the ANS, it is difficult to imagine that these two systems are in no way correlated. However, *how* they interact and are related over time is still largely unknown and this review provides a first step in disentangling their relationship.

## Supporting information

**S1 File. Example of a search strategy.**
(DOCX)

**S2 File. Study population characteristics per cardiovascular measures for stress.**
(DOCX)

**S3 File. Screening phases.**
(XLSX)

**S4 File. Extraction data from retrieved studies.**
(XLSX)

**S1 Table. Associations based on study methods.**
(DOCX)

**S2 Table. Methodological quality of the included studies.**
(DOCX)

**S1 Checklist. PRISMA 2009 checklist.**
(DOC)

## Author Contributions

**Conceptualization:** Thomas Vaessen, Aki Rintala, Wolfgang Viechtbauer.

**Data curation:** Thomas Vaessen, Aki Rintala.

**Formal analysis:** Thomas Vaessen, Aki Rintala.

**Funding acquisition:** Stephan Claes, Inez Myin-Germeys.

**Investigation:** Thomas Vaessen, Aki Rintala.

**Methodology:** Thomas Vaessen, Aki Rintala, Natalya Otsabryk.

**Project administration:** Thomas Vaessen, Aki Rintala.

**Resources:** Thomas Vaessen, Aki Rintala.

**Software:** Thomas Vaessen, Aki Rintala.

**Supervision:** Wolfgang Viechtbauer, Martien Wampers, Stephan Claes, Inez Myin-Germeys.

**Validation:** Thomas Vaessen, Aki Rintala.

**Visualization:** Thomas Vaessen, Aki Rintala.

**Writing – original draft:** Thomas Vaessen, Aki Rintala, Inez Myin-Germeys.

**Writing – review & editing:** Thomas Vaessen, Aki Rintala, Wolfgang Viechtbauer, Martien Wampers, Stephan Claes, Inez Myin-Germeys.

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
