## [Decision Letter · Decision Letter 0]

27 May 2021

PONE-D-20-29015

The association between self-reported stress and cardiovascular measures in daily life: A systematic review

PLOS ONE

Dear Dr. Vaessen,

Thank you for submitting your manuscript to PLOS ONE. After careful consideration, we feel that it has merit but does not fully meet PLOS ONE’s publication criteria as it currently stands. Therefore, we invite you to submit a revised version of the manuscript that addresses the points raised during the review process.

We look forward to receiving your revised manuscript.

Kind regards,

Ricarda Nater-Mewes, PhD

Academic Editor

PLOS ONE

Journal Requirements:

3. The PRISMA checklist requires that the quality or risk of bias of the included studies is assessed. Furthermore, conclusions in a systematic review or meta-analysis should be related to the quality of the included publications. In light of this concern we would be grateful if you could please update your systematic review to include a formal quality assessment. Please ensure that your quality assessment is suitable for the study type included in your systematic review. Please also note that reporting checklists (e.g. STROBE) are not suitable for use as quality assessment tools.

5. Please include captions for ALL your Supporting Information files at the end of your manuscript, and update any in-text citations to match accordingly. Please see our Supporting Information guidelines for more information: http://journals.plos.org/plosone/s/supporting-information.

Reviewers' comments:

Reviewer's Responses to Questions

**Comments to the Author**

1. Is the manuscript technically sound, and do the data support the conclusions?

Reviewer #1: Yes

Reviewer #2: Yes

2. Has the statistical analysis been performed appropriately and rigorously? 

Reviewer #1: N/A

Reviewer #2: Yes

3. Have the authors made all data underlying the findings in their manuscript fully available?

Reviewer #1: No

Reviewer #2: Yes

4. Is the manuscript presented in an intelligible fashion and written in standard English?

Reviewer #1: Yes

Reviewer #2: Yes

5. Review Comments to the Author

Reviewer #1: This is an interesting and relevant overview of the literature on associations between self-reported stress and selected cardiovascular measures. As with all reviews, the outcome is dependent on a variety of choices made by the authors – some might be debatable (such as the focus on at least one day of assessments), but as long as these choices are justified, the end result stands on a relatively firm basis. I have only a few suggestions on how to further improve the paper:

- The main query I have is that that I would want to ask the authors to put together a brief list of recommendations at the end of their discussion. I think they provide a clear-cut discussion on their findings, and it becomes quite clear what the limitations in the literature are, but at the end the reader is left wondering what to take from the paper for his or her own research. What is, e.g., an ideal segment for the association between self-reported stress and HRV? What is the ideal statistical approach based on which data characteristics? What is the ideal time-lag between self-reported stress and selected measures? Just a few questions I had.

- The main variables in this review should probably be justified a little more in detail. The authors start with an overview of the biological stress systems, but I don´t think this is of great relevance, when in the end they are only focusing on a few selected cardiovascular measures. They could simply skip to the role of the cardiovascular system and its role in the overall stress process. In this context, it should be noted that there are literally hundreds of ESM studies which measured cortisol, so the statement on page 4 regarding HPA axis markers is not correct.

- I cannot agree that laboratory studies are showing heterogeneous results due to a lack of ecological validity. If one wants to follow the stress appraisal model by Lazarus (and most studies seem to do to some degree), then it doesn´t matter where the stressor takes place. The authors could leave out that statement and still have enough to justify their focus on ESM studies.

- I agree that it is difficult to compare studies due to the wide variety of statistical approaches. One of the conclusions could be that it will be of importance to follow-up with a meta-analysis, which would allow for tackling the main research question using one unified statistical approach.

Reviewer #2: This systematic review evaluated the evidence for studies that have assessed daily stress using ambulatory methods and associated ratings of self-reported stress with cardiovascular measures. The most interesting point is that it focuses on daily stress. I agree that the results from laboratory setting studies using standardized stress task are different from those measured in the real-life stressful condition. This paper is interesting throughout, but I have a few minor comments.

1. One of the aims of this review is to investigate an association between acute stress and physiological measures. It would be interesting if the analysis could be limited to only those studies that assessed acute stress.

2. P33 line 3: “One possible explanation for sample differences is that patients and individuals at risk may experience more stress during the day, providing for more variability in the data and hence increase the likelihood of finding an association.” Does this mean that the patients and individuals at risk are more frequently exposed to chronic stress, or does it mean that they are more sensitive to stressors?

3. P34 line2: Validation of BP monitoring device evaluates the accuracy of BP values measured by BP monitoring devices compared with those measured by mercury sphygmomanometers, that is, Spacelabs 90207 is a validated device for monitoring BP.

6. PLOS authors have the option to publish the peer review history of their article (what does this mean?). If published, this will include your full peer review and any attached files.

Reviewer #1: **Yes: **Urs Nater

Reviewer #2: No

---

## [Author Response · Author response to Decision Letter 0]

7 Jul 2021

Comments from the reviewers:

Reviewer #1: This is an interesting and relevant overview of the literature on associations between self-reported stress and selected cardiovascular measures. As with all reviews, the outcome is dependent on a variety of choices made by the authors – some might be debatable (such as the focus on at least one day of assessments), but as long as these choices are justified, the end result stands on a relatively firm basis. I have only a few suggestions on how to further improve the paper:

1. The main query I have is that that I would want to ask the authors to put together a brief list of recommendations at the end of their discussion. I think they provide a clear-cut discussion on their findings, and it becomes quite clear what the limitations in the literature are, but at the end the reader is left wondering what to take from the paper for his or her own research. What is, e.g., an ideal segment for the association between self-reported stress and HRV? What is the ideal statistical approach based on which data characteristics? What is the ideal time-lag between self-reported stress and selected measures? Just a few questions I had.

Response: This is a very good suggestion. Although it is difficult to provide strong suggestions for all points the referee suggests based on the results of the current review, we have changed the title of the final paragraph to read “Conclusions and recommendations” and added some recommendations based on our findings. This can be found on page 35.

“Therefore, based on this review, we recommend researchers to use the following when investigating the association between self-reported stress and cardiovascular measures:

- Perceived stress or high-arousal NA self-report measures

- High-quality wearable sensors that have been validated in ambulatory settings

- Time windows of max 30 minutes prior to self-report for the calculation of continuously assessed cardiovascular measures

- Study periods of at least 6 days with multiple assessments per day to capture enough variability in both measures

- Multilevel modelling for the statistical analyses”

2. The main variables in this review should probably be justified a little more in detail. The authors start with an overview of the biological stress systems, but I don´t think this is of great relevance, when in the end they are only focusing on a few selected cardiovascular measures. They could simply skip to the role of the cardiovascular system and its role in the overall stress process. In this context, it should be noted that there are literally hundreds of ESM studies which measured cortisol, so the statement on page 4 regarding HPA axis markers is not correct.

Response: Although we do agree with the referee that it is unnecessary to mention the HPA axis in the introduction, our intention was to review studies investigating ANS measures (ie including skin conductance). As we only found out after our search that there were not enough studies that looked at the association between skin conductance and self-reported stress, we chose to focus on cardiovascular measures only. In our opinion, altering the introduction accordingly would not be in line with our procedures. We therefore deleted the two sentences on the HPA axis on page 4, but left the motivation to investigate the ANS intact. We hope the referee agrees with this decision.

3. I cannot agree that laboratory studies are showing heterogeneous results due to a lack of ecological validity. If one wants to follow the stress appraisal model by Lazarus (and most studies seem to do to some degree), then it doesn´t matter where the stressor takes place. The authors could leave out that statement and still have enough to justify their focus on ESM studies.

Response: We fully agree with the referee that the stress system does not differentiate between a laboratory stress task or a daily stressor. However, depending on the nature of the stressor, we do believe the stress response differs between different instances. The very abrupt on- and offset of stressor exposure in the lab, where often anticipatory stress is tried to be kept to a minimum by inclusion of a baseline period, is arguably not comparable to typical daily-life stressors such as work stress, social stressors, ruminative thoughts, etc. that have much less clear start- and endpoints. In that sense, the stress response that we typically observe in the lab probably does not reflect stress system activity in daily life. However, we do agree with the referee that we cannot conclude that this is the reason for heterogeneous results from lab tasks and have therefore deleted this sentence on page 5.

4. I agree that it is difficult to compare studies due to the wide variety of statistical approaches. One of the conclusions could be that it will be of importance to follow-up with a meta-analysis, which would allow for tackling the main research question using one unified statistical approach.

Response: This is a very good suggestion and we have added the following sentence to our discussion on page 34.

“Although most studies used MLM, the large diversity between models prevented us from conducting meta-analyses based on the reported values of the associations per stress measure. Therefore, an accurate estimate of the strength of the association could not be provided. Future investigations should put in effort to mirror the statistical models and methods previously used to estimate the association between self-reported and cardiovascular measures of stress so that meta-analyses can be conducted.”

Reviewer #2: This systematic review evaluated the evidence for studies that have assessed daily stress using ambulatory methods and associated ratings of self-reported stress with cardiovascular measures. The most interesting point is that it focuses on daily stress. I agree that the results from laboratory setting studies using standardized stress task are different from those measured in the real-life stressful condition. This paper is interesting throughout, but I have a few minor comments.

1. One of the aims of this review is to investigate an association between acute stress and physiological measures. It would be interesting if the analysis could be limited to only those studies that assessed acute stress.

Response: We agree with the referee that looking at associations for acute stress only would be very interesting. However, it is, using these data, impossible to disentangle acute from chronic stress. Therefore, to prevent confusion, we have removed the word “acute” from our aims paragraph on page 5.

2. P33 line 3: “One possible explanation for sample differences is that patients and individuals at risk may experience more stress during the day, providing for more variability in the data and hence increase the likelihood of finding an association.” Does this mean that the patients and individuals at risk are more frequently exposed to chronic stress, or does it mean that they are more sensitive to stressors?

Response: Good point. We meant overall more fluctuations in their stress levels due to both more exposure to stressors and increased sensitivity to stress, as only higher overall levels of stress would not result in more fluctuations throughout the day. We have clarified this as follows:

“One possible explanation for sample differences is that patients and individuals at risk may experience more fluctuations in their stress levels during the day, providing for more variability in the data and hence increase the likelihood of finding an association.”

3. P34 line2: Validation of BP monitoring device evaluates the accuracy of BP values measured by BP monitoring devices compared with those measured by mercury sphygmomanometers, that is, Spacelabs 90207 is a validated device for monitoring BP.

Response: We agree with the reviewer. However, we also believe that we have made this point very clear in our manuscript. In the line the referee refers to, we explicitly state that Spacelabs 90207 indeed is a valid tool for BP monitoring. We do feel that we have to mention the limitations that are described in the literature regarding the device. We hope the referee agrees that our current phrasing accurately describes both.

---

## [Decision Letter · Decision Letter 1]

22 Oct 2021

The association between self-reported stress and cardiovascular measures in daily life: A systematic review

PONE-D-20-29015R1

Dear Dr. Vaessen,

We’re pleased to inform you that your manuscript has been judged scientifically suitable for publication and will be formally accepted for publication once it meets all outstanding technical requirements.

Kind regards,

Ricarda Nater-Mewes, PhD

Academic Editor

PLOS ONE

Reviewers' comments:

Reviewer's Responses to Questions

**Comments to the Author**

1. If the authors have adequately addressed your comments raised in a previous round of review and you feel that this manuscript is now acceptable for publication, you may indicate that here to bypass the “Comments to the Author” section, enter your conflict of interest statement in the “Confidential to Editor” section, and submit your "Accept" recommendation.

Reviewer #1: All comments have been addressed

Reviewer #2: All comments have been addressed

2. Is the manuscript technically sound, and do the data support the conclusions?

Reviewer #1: (No Response)

Reviewer #2: Yes

3. Has the statistical analysis been performed appropriately and rigorously? 

Reviewer #1: (No Response)

Reviewer #2: Yes

4. Have the authors made all data underlying the findings in their manuscript fully available?

Reviewer #1: (No Response)

Reviewer #2: Yes

5. Is the manuscript presented in an intelligible fashion and written in standard English?

Reviewer #1: (No Response)

Reviewer #2: Yes

6. Review Comments to the Author

Reviewer #1: (No Response)

Reviewer #2: The manuscript is well revised for the points commented by reviewers.

Compared to the previous version, this revised manuscript is more systematically organized and easier to understand.

I have no further comments.

7. PLOS authors have the option to publish the peer review history of their article (what does this mean?). If published, this will include your full peer review and any attached files.

Reviewer #1: **Yes: **Urs M. Nater

Reviewer #2: No

---

## [Editor Report · Acceptance letter]

3 Nov 2021

PONE-D-20-29015R1 

The association between self-reported stress and cardiovascular measures in daily life: A systematic review 

Dear Dr. Vaessen:

I'm pleased to inform you that your manuscript has been deemed suitable for publication in PLOS ONE. Congratulations! Your manuscript is now with our production department. 

Kind regards, 

on behalf of

Dr. Ricarda Nater-Mewes 

Academic Editor

PLOS ONE